# Energy requirements for securing wellbeing in Switzerland and the space for affluence and inequality

Joel Millward-Hopkins [1] ✉, Vivien Fisch-Romito [1], Sascha Nick [2] & Emile Chevrel[3]

The idea that human needs should be secured for all people is largely uncontroversial, and recent research demonstrates that decent living standards could be secured for all, globally, with far lower energy and resource use than today. However, how the energy requirements of decent living vary across populations is poorly understood – particularly in high-income countries—and important questions regarding inequality remain unexplored. Here we show how, with a fairer distribution of energy, Switzerland could dramatically reduce energy consumption while securing wellbeing for all. We advance previous work on energy and wellbeing by decomposing an established net-zero scenario into the energy required to support human needs, and that related to affluence or excess. We estimate decent living energy in 2050 at 19.5 gigajoules per capita (18–26 gigajoules in varying subnational contexts), making it only ~13% of Switzerland's 2019 energy footprint, and ~23% of that projected in the net-zero scenario. This highlights the theoretical potential for affluent countries to move towards a more just, egalitarian global distribution of energy and resource consumption, while securing wellbeing for their own citizens.

The question of how human wellbeing can be secured with a minimum amount of energy use is receiving a growing amount of attention. Building upon the concept of Decent Living Standards[1,2], researchers have estimated that the material prerequisites needed to secure human wellbeing could be provided with only ~10–30 gigajoules (GJ) per capita per year of final energy[3–6] – a fraction of current global final energy consumption[4]. Other research suggests these same living standards could be met universally without breaching planetary boundaries[7] and with reduced global material consumption[8]. These findings are made more dramatic by considering that the majority of the world's population currently fall short of multiple aspects of decent living, most commonly mobility, cooling, and sanitation[6].

These studies, along with closely related work developing Low Energy Demand scenarios[9,10], offer encouraging news. During the past decade, global carbon emissions and anthropogenic warming

rates reached unprecedented levels[11], despite 30 years of climate change talks attended by political leaders. Current levels of global emissions may exhaust a 1.5 °C budget in just a few years (*ibid*). Furthermore, many scenarios keeping warming to safe levels rely upon potentially unrealistic and infeasible technological deployment, most notably a grand scale of carbon capture[12], which presents issues for both biodiversity[13] and equity[14]. Futures in which energy demand can be substantially reduced—without harming wellbeing outcomes in affluent countries (particularly the World Bank High-Income countries and many of their Upper-middle Income countries) and improving them in poorer regions—thus offer ways forward that mitigate the transition risk of technologically-demanding mainstream scenarios[15]. Such demand reduction is also appealing from a national perspective, as it makes energy security (or even self-sufficiency) much easier to achieve.

[1]University of Lausanne (UNIL), Quartier Centre, Lausanne, Switzerland. [2]Swiss Federal Technology Institute of Lausanne (EPFL), Lausanne, Switzerland. [3]ETH Zurich, Zurich, Switzerland. ✉e-mail: joeltmh@gmail.com

However, research estimating the energy requirements of providing decent living standards is relatively young, and how these requirements may vary in different contexts remains poorly understood. This is particularly true in the global North, as work has often focused on the global South[3] and the considerable decent living gaps that exist there[6]. Models with significant global coverage have taken national contexts into account in limited ways, and submodels of mobility have lacked any empirical support[5]. Further, some consumption sectors essential to wellbeing remain absent and the question of inequality is often overlooked entirely[16,17].

Here we expand the existing body of work on energy and wellbeing by assessing the minimum energy use required to provide wellbeing in Switzerland and how this may vary in different subnational contexts. We then explore the space that remains for energy inequality—the magnitude of energy inequality that could exist (regardless of its desirability), given the minimum energy required for wellbeing, and the limits that global sustainability objectives imply. Switzerland presents an interesting case—globally, it has one of the cleanest national electricity grids, but among the highest per-capita carbon footprints[18]. Although climate policy in Switzerland has faced challenges in recent years[19], detailed technology pathways to reduce and shift energy use to meet net-zero in 2050 have been developed by the Swiss Federal Office of Energy[20]. However, as we describe later, these focus on energy use within Switzerland only, and the high global energy use required to support the (high) Swiss consumption these pathways assume is likely either unsustainable or unjust[21]. It is thus useful to explore how much further energy use could be reduced without compromising wellbeing.

We use a standard bottom-up modelling approach to make a first estimate of Decent Living Energy (DLE) for Switzerland. DLE models estimate the energy requirements of providing the goods and services that form Decent Living Standards (DLS; Table 1) – living standards considered prerequisites for securing human wellbeing. Existing models work with final energy, as this is a step closer than primary energy to the goods and services that support human wellbeing[5]. And instead of territorial measures of energy use, they work with energy footprints, which, like carbon footprints, include both the direct and indirect (i.e. embodied) impacts associated with all goods and services consumed by a population.

Conceptually, DLE models are simple, with two core ingredients: The first is activity levels, quantitative indicators of consumption across all the dimensions of the decent living standards inventory. For example, for mobility, annual activity-levels must be specified as passenger-kilometres (pkm) travelled per person; for shelter, metres residential floor-space of a household (m²); for clothing, annual consumption of new clothing in kilograms (kg) per person. Our activity levels (Table 1) are a combination of general normative assumptions made in previous global models, and bespoke estimates for Switzerland – most importantly, for mobility. The second ingredient is energy intensities for each activity, for example, the intensities of different transport modes (MJ pkm⁻¹), residential thermal comfort (MJ m⁻²), and clothing production (MJ kg⁻¹). All our energy intensities are, in contrast to previous models, Swiss-specific and underpinned largely by the net-zero pathway mentioned above[20]. Full details on the derivation of DLS activity levels and energy intensities are included in the Methods (and Supplementary Methods 1 and 2).

We find that, on a net-zero emissions pathway for Switzerland, only ~20% of the projected energy use is required to support decent living standards for all. Theoretically, this suggests that, within this future, the lowest Swiss consumers could access sufficient energy consumption, even if Swiss energy inequality increased. However, an alternative interpretation it that this highlights the potential for Switzerland to reduce energy use significantly more, while still securing wellbeing for its own citizens. The average Swiss energy footprint in 2050 could potentially be reduced to the global average per-capita

## Table 1 | List of decent living standard dimensions, with those added for the current work indicated by asterisks

| High-level category | Subcategory | Key activity-levels |
|---|---|---|
| Nutrition | Food<br>Cooking appliances & cold storage | 2557 kcal per cap per day⁺ |
| Shelter & living conditions | Sufficient space & thermal comfort<br>Household furnishings*<br>Illumination | 20 m² per cap⁺ |
| Hygiene | Water supply<br>Water heating<br>Cleaning products*<br>Waste management | 50 L per cap per day (20 L heated) of direct water use in the home |
| Clothing | Clothes<br>Washing facilities | 3.6 kg per cap per year |
| Healthcare | Hospitals | See Supplementary Methods 1 |
| Education | Schools | 10 m² per pupil |
| Communication & information | Phones & Computers<br>Networks + data centres | See Supplementary Methods 1 |
| Mobility | Vehicle manufacture<br>Vehicle propulsion<br>Transport infrastructure | ~4000–9000 pkm per cap per year⁺ |
| Public space* | Buildings for leisure, art, culture, etc.* | 1.6 m² per cap |
| Other public activities* | Administration, security, research, etc.* | See Supplementary Methods 1⁺ |
| Unspecified | Retail services, power infrastructure, freight. | See Supplementary Methods 1⁺ |

Key activity-levels are listed in the rightmost column, with all Swiss-specific estimates indicated with crosses. Note this is not an exhaustive account of activity-levels – further details are in Supplementary Methods 1, along with descriptions of how these levels are derived.

levels found in IPCC net-zero scenarios consistent with 1.5–2 °C of warming and low negative emissions dependency.

## Results

### Future Swiss energy use

Current final energy consumption in Switzerland is nearly 90 GJ cap⁻¹ year⁻¹, from a domestic (territorial) perspective. The Net-Zero Basis (NZB) scenario of Energy Perspectives 2050 (EP2050+)[20] describes a pathway for domestic Swiss energy use that achieves carbon neutrality in 2050 by assuming rapid deployment of clean and efficient technologies across all sectors and utilisation of carbon capture, but a continuation of current Swiss living standards and no reductions in energy service demands. It projects Swiss energy use to fall to 51 GJ cap⁻¹ in 2050 (Fig. 1a), thus slightly lower than the 60 GJ cap⁻¹ projected in business-as-usual (BAU) in the same report. For carbon emissions, differences between these scenarios are much more significant, with BAU projecting a ~30% decrease by 2050 and NZB projecting 100% (by definition).

The final energy footprint of Switzerland is much higher than this domestic consumption. We derive values from EXIOBASE of 147 GJ cap⁻¹ in 2019 (prior to COVID) and 179 GJ cap⁻¹ in 2012[22], so ~70% larger than domestic consumption (Fig. 1b). The footprint includes direct and indirect final energy use supporting Swiss consumption from households, governments, non-profits, and infrastructure formation, but not the energy use of commercial Swiss activities that produce for export. Other Swiss research has reported similar values, including that the 2011 final energy footprint was 80% larger than domestic consumption[23].

It is not clear whether this current ratio of energy footprint to domestic energy in Switzerland would decrease or increase in the NZB

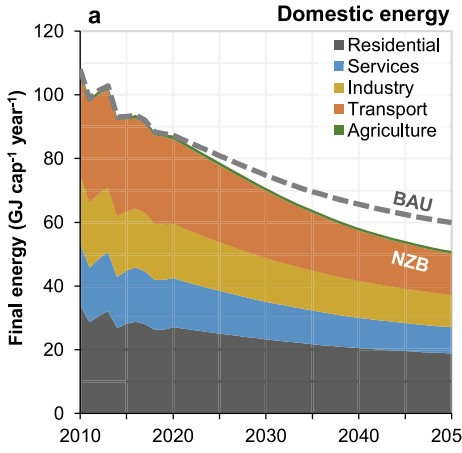

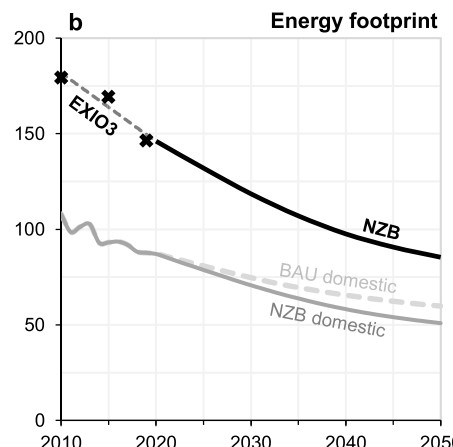

**Fig. 1 | Baseline energy consumption of Switzerland. a** Domestic energy use in the EP2050+ Net Zero Basis (NZB) scenario disaggregated into five sectors, alongside the EP2050+ business-as-usual (BAU) scenario for reference. **b** Switzerland's full energy footprint, including both historical EXIOBASE3 data and our upscaling of the EP2050+ NZB scenario (grey lines on **b** indicate the same domestic-scope data from **a**). Note, these figures are reproduced from Swiss FOE supplementary data[20] and EXIOBASE, so should not be understood as our results.

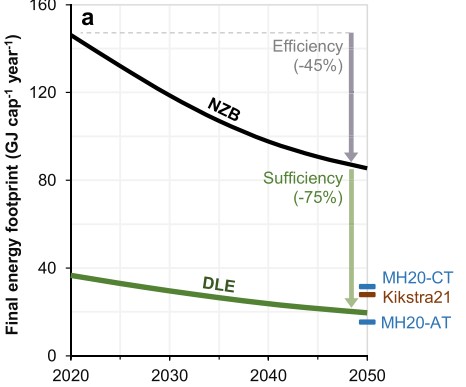

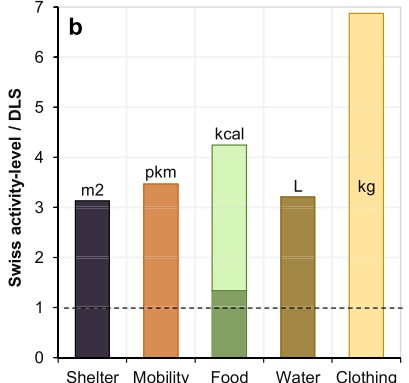

**Fig. 2 | Decent living energy and normalised activity-levels. a** The Net Zero Basis (NZB) scenario and our Decent Living Energy (DLE) estimate, with the resulting energy use reductions decomposed into efficiency and sufficiency. Also shown are comparable DLE estimates from existing global models: the current technology (MH20-CT) and advanced technology (MH20-AT) scenarios from Millward-Hopkins et al.[5] for Switzerland, and data from Kikstra et al.[6] for Western Europe (Kikstra21). **b** Current consumption levels in Switzerland relative to our assumed Decent Living Standard (DLS) activity levels from Table 1. For food, the darker bar indicates overconsumption of all food, and the lighter bar overconsumption of animal-based foods in particular. Note, these ratios largely arise from our assumptions and are not results, with the exception of mobility as our assumed activity-levels arises from the analysis of trips and service accessibility, which is described in Supplementary Methods 1.

pathway of EP2050+. On the one hand, EP2050+ aims for national energy security, thus eliminating the Swiss energy footprint currently embodied in imported fossil fuels. On the other, producing the technologies that a net-zero future relies upon – imported electric cars, solar panels, etc. – will require significant amounts of industrial energy use outside of Switzerland. Further, the strong domestic energy efficiency improvements that EP2050+ assumes may not be matched in the global supply chains that Switzerland will continue to rely upon for things like household appliances and medical supplies, which could increase the share of the Swiss footprint that is imported.

We thus produce an adjusted-NZB scenario to reflect the full Swiss energy footprint, by assuming that the domestic share of this footprint remains unchanged from its 2019 value (Fig. 1b). Accordingly, the energy intensities in our DLE model are time-dependent, as they are primarily derived from the sectoral data underlying this this net-zero pathway (see Methods for further details).

### Decent living energy

We estimate DLE in Switzerland to be ~37 GJ cap⁻¹ in 2020, falling to 19.5 GJ cap⁻¹ in 2050. This implies that 25% of the Swiss energy footprint currently supports decent living standards, with this falling marginally to 23% in 2050. The remaining ~75% is related either to affluence or inefficient means of meeting needs. Our 2050 estimate is close to the values reported in global work for Switzerland[5] and Western Europe[6] (Fig. 2a).

We also consider the energy reductions that are due to sufficiency and efficiency, by comparing the difference between (i) the 2020 (or 2050) Swiss energy footprint and DLE estimate (sufficiency) and (ii) the 2020 and 2050 NZB (or DLE) estimates, which is due to the technological improvements assumed in EP2050+. The former is an ~75% reduction, the latter a ~45% reduction, implying sufficiency is much more significant than efficiency. Sufficiency is so effective because current activity-levels in Switzerland are ~3–7 times higher than decent living standards across key sectors (Table 1 and Fig. 2b). However, note that current Swiss energy use is the average of unequal use across society, while our DLE estimate is a floor below which no one should fall, not a level that the whole of Switzerland can be expected to reach. This should be borne in mind for the above comparison, and we return to the question of inequality below.

### Composition and variation of decent living energy

Existing DLE models typically focus on national- or regional-level analysis. Similarly, the basic DLE results of the previous section were a national aggregate across all people, assuming the average mix of

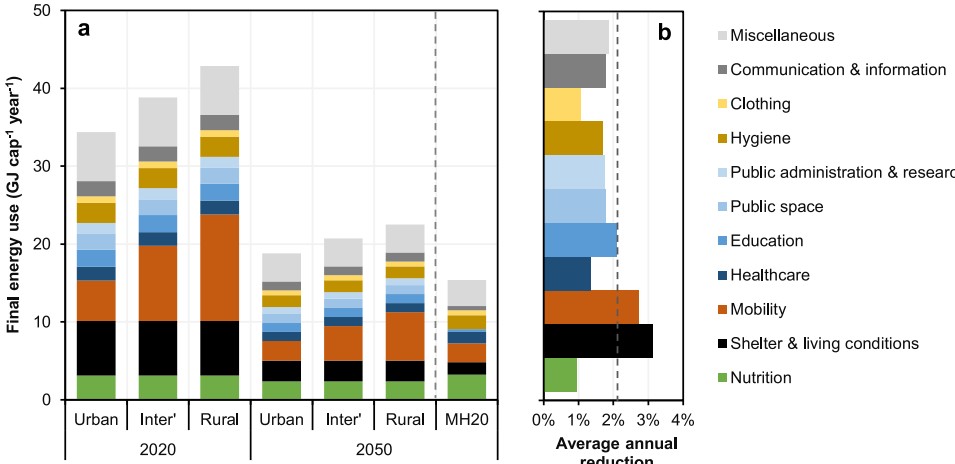

**Fig. 3 | Decent living energy by geography and its temporal change. a** Decent Living Energy (DLE) for the three geographies considered (inter' = intermediate), shown for 2020 and 2050 and broken down by the Decent Living Standards dimensions of Table 1. Shown for comparison is the DLE estimate for Switzerland in 2050 from Millward-Hopkins et al.[5], assuming advanced technologies (MH20). **b** The average annual energy use reduction for each consumption-sector over the 2020–2050 period, specifically the Compound Annual Growth Rate, with the dotted line indicating the average for the total footprint.

**Table 2 | Descriptions of three scenarios for which we calculate variations in Decent Living Energy**

| Factor | Low-energy case | High-energy case | Central case |
|---|---|---|---|
| Mobility pattern | Person assumed to be living car-free. All pkm by car in the central case is shifted to bus and rail, in proportion to the existing shares of these modes). | Person assumed to have high car dependence (due to health conditions, occupational requirements, and/or other special circumstance). Walking, cycling, and bus pkm in the central case is shifted to car travel, but overall distance is not changed. | Mode share from our DLS mobility model (see Supplementary Methods 1). |
| Car propulsion | Person assumed to have access to an electric car. | Person assumed to only have access to an older gasoline car. The energy intensity of a 2030 gasoline vehicle is used for the 2050 DLE projection. | Average of the vehicle fleet composition assumed in EP2050+ (98% fossil fuel based in 2020, falling to under 15% in 2050). |
| Residential heating system | Person assumed to live in a household with a heat pump. | Person assumed to live in a household with a standard gas boiler. | Average of the housing stock heating systems composition assumed in EP2050+ (heat pumps and district heating increasing from 25% of stock in 2020 to 90% in 2050). |

Abbreviations are as follows: *DLS* Decent Living Standards, *DLE* Decent Living Energy, *EP2050+* Energy Perspectives 2050+, pkm. passenger kilometres.

technologies in Switzerland. In practice, numerous contextual factors influence the level of consumption in different DLS categories that is required to satisfy needs: those with more caring responsibilities may require higher levels and different forms of mobility; those with higher metabolisms require more food; those suffering from chronic health conditions require more frequent and intensive healthcare; older people may require warmer homes[24]. DLE varies further with the type and efficiency of end-user technologies utilised by individuals and households, which may be constrained by affordability or availability. Expanding a DLE model to consider all significant contextual and demographic factors would be impractical, and unnecessary for the high-level picture we explore in this work. We thus limit analysis to consider three key geographies, and three separate contextual factors. First, we consider geography, while also considering the energy required for different aspects of decent living (see Table 1).

Figure 3 shows DLE separately for Urban, Intermediate, and Rural populations, as defined by the Swiss Federal Statistical Office. Differences here are entirely due to mobility-related assumptions, which lead to assumed activity levels in rural areas being over double those in urban areas (see Supplementary Table 5) with a mode share shifted towards cars. In 2020, this results in DLE ranging from 34 to 43 GJ cap⁻¹, while in 2050 the spread reduces to 19–23 GJ cap⁻¹. As the urban population is dominant (63% in 2020 and 69% in 2050), nationally averaged DLE is close to the bottom of these ranges. One can also see

from Fig. 3 that mobility and shelter & living conditions are the most significant DLS dimensions contributing to DLE in 2020, with the latter reducing in importance by 2050 as residential heating improves. Also notable here is that the categories absent from previous DLE work – public space, other public activities, household furnishings, cleaning products – account for 5 GJ cap⁻¹ in 2020 and 3 GJ cap⁻¹ in 2050, a significant ~14% of total DLE. In contrast, critical sectors such as healthcare and education together account for only ~11%.

We then consider three further contextual factors as a partial insight into how DLE may vary across a heterogenous population. For each factor, we consider a case of both high and low energy consumption (Table 2). No single factor has a transformative impact on total DLE, never increasing or reducing it by more than ~10% from the central case (Fig. 4). However, it is significant that the total energy footprint needed to provide a person with DLS can be changed this much by a single technology. Relative to the high case of gas heating systems, residential heat pumps may reduce total DLE by ~15%. Relative to gasoline cars, electric cars may reduce DLE of rural inhabitants by the same percentage in 2050. In contrast, it is perhaps surprising that the variation in DLE between contexts of high- and zero-car use is not larger. The influence on DLE for a typical rural inhabitant is considerable – in 2020, the low case of zero-car use decreases total DLE by 19% relative to high-car use. But by 2050, this reduction falls to 11%, and for typical urban inhabitants it is only 6%.

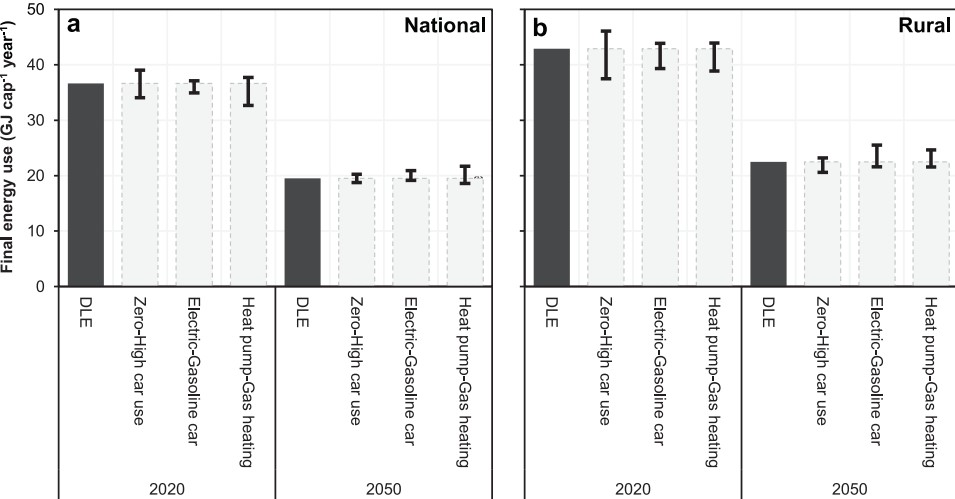

**Fig. 4 | Contextual factors influencing decent living energy.** Total Decent Living Energy (DLE) in 2020 and 2050, and the effect of the three contextual factors described in Table 2. Data is shown averaged across Switzerland (**a**), and for rural areas (**b**). Centre lines indicate the range in DLE between the low and high estimates describe in the main text – they are not box-and-whisker plots representing statistical data.

There appear to be three reasons for this. First, direct energy for car transport in 2050 (averaged across urban, intermediate and rural areas) only accounts for ~1 GJ cap⁻¹ (5%) of total DLE, largely due to the assumed ~60% reduction in total mobility in the DLE scenario, relative to current levels. So even eliminating the direct energy use of car transport entirely has only a modest impact on DLE. In other words, once total passenger mobility levels are reduced to sufficiency levels, recomposing the mode share of the remaining travel has much less importance for energy consumption. Second, EP2050+ assumes much greater efficiency gains over the 2020–2050 period for private vehicles (-55%) than for rail (-25%) and bus transport (-30%). So by 2050, the direct energy intensity of car travel is only projected to be just over twice that of rail. Third, the indirect energy consumption per pkm we calculate for car and rail transport is relatively similar, thus bringing the total energy use per pkm in 2050 even closer. We return to these issues in the conclusions.

### Sustainable energy footprints and energy inequality

DLE represents an energy floor for the lowest Swiss consumers. We can use this to explore the space available for inequality after also estimating: (i) the average sustainable amount of energy available in Switzerland per-person and (ii) the shape of the energy footprint distribution.

Regarding (i), although EP2050+ charts a pathway that reaches net-zero, its focus on domestic energy use only means it cannot tell us if the associated (global) Swiss energy footprint is sustainable. We thus look to global estimates of sustainable energy use. In the context of climate mitigation, it is common to take a simple egalitarian approach, which distributes global carbon budgets between countries in proportion to population, thus allocating every human an equal amount. Here we apply the same egalitarianism to energy[21]. DeAngelo, et al.[25] summarise how global average final energy use (per capita per year, in the mid-late century) varies across IPCC scenarios meeting different temperature goals, including many scenarios relying on high levels of negative emissions. They report a range of 24.5–76.2 GJ (mean of 48.3 GJ) in 1.5 °C scenarios, and 36.1–88.7 GJ (mean of 61.2 GJ) for 2 °C scenarios, around mid-century. The mean of 1.5 °C scenarios is thus very close to what the EP2050+ NZB scenario projects for Switzerland's domestic energy use in 2050 (51 GJ). However, the Swiss footprint implicit to EP2050+ NZB (85 GJ cap⁻¹ in 2050) is at the upper end of the 2 °C range. Further, NZB assumes substantial deployment of CCS (40% occurring outside Switzerland), with this reaching 1.2 t CO₂e cap⁻¹ in

2050 – a level that, if extended globally, would likely encounter feasibility constraints[12]. These observations present a case for reducing Swiss energy use further than EP2050+ NZB assumes.

Regarding (ii), to model inequality, we use the analytical formulas recently provided by Pauliuk[26]. These provide a simple way to relate DLE and the Gini coefficient (independent variables) to the mean consumption of a population or to the consumption of the top decile (dependent variables). Note, Gini coefficients range from 0 to 1, where 0 represents perfect equality. The output of these formulas is shown in Fig. 5a, and the shaded area thus shows the assumed range of energy consumption across the full Swiss population, for different Gini coefficients. We estimate the current energy Gini for Switzerland to be 0.27 – at this level of inequality, if the bottom decile consumed DLE, mean consumption would be 1.7 × DLE and maximum consumption 4.6 × DLE. Further details on all calculations are described in the Methods.

### The space available for energy inequality

Figure 5b shows how increasing levels of energy inequality require increases in average 2050 Swiss energy use in order to secure DLE for the lowest consumers. The ranges of global average energy use compatible with 1.5 °C and 2 °C from DeAngelo et al.[25] are indicated, and curves for three different DLE thresholds – the 19.5 GJ cap⁻¹ year⁻¹ of the current work, and the lowest and highest thresholds from previous global models[4–6]. Finally, current energy inequality is shown, and an approximation of fair inequality. We can see that if Switzerland's energy footprint in 2050 was around the mean value of DeAngelo, et al.[25] for 1.5 °C or 2 °C futures (~50–60 GJ, so well below the NZB scenario), then there would be substantial room for energy inequality (Fig. 5b). We estimate the Gini could increase to 0.4–0.5 before the lowest Swiss consumers were pushed below DLE of 19.5 GJ cap⁻¹ year⁻¹. This implies that, even without reducing existing inequalities, Switzerland could bring its energy footprint down to a global average consistent with the mean of 1.5–2 °C scenarios, while providing all citizens with energy use above DLE. But there exist several reasons to be cautious about this conclusion.

One is that the space available for inequality is highly dependent upon the level of energy use considered sustainable. The DeAngelo, et al.[25] ranges are broad and there are substantial concerns regarding the feasibility of the quantities of negative emissions deployed in many of the underlying scenarios[12]. It is thus likely that technically realistic sustainable global energy budgets lie towards the lower end of the

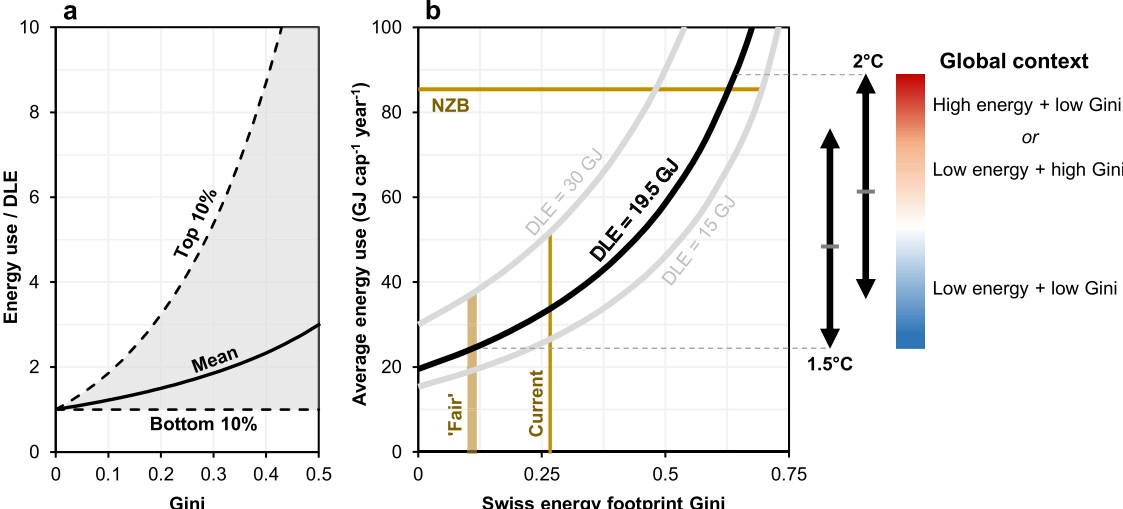

**Fig. 5 | Decent living energy and the space for inequality. a** The relationship between a given Decent Living Energy (DLE) floor, and the average and maximum consumption of a population, according to the equations of Pauliuk[26]. All data here is normalised by DLE to give dimensionless curves. Note that panel a is not results, but rather it is offered to aid understanding of the right plot. **b** Relationships between the maximum feasible energy footprint Gini in Switzerland and the average energy footprint, in 2050. Feasible here means the Gini at which point the lowest consumers are placed at the DLE floor, for a given average level of Swiss energy use (as indicated on the Y-axis). Lines for three different DLE floors are shown, as well as an indication of current and fair inequality (see Methods for details). Ranges of energy use consistent with different temperature goals are from DeAngelo, et al.[25]. The colour bar illustrates approximately what global context is consistent with different levels of average Swiss energy use.

scenario ranges. Additionally, for the lowest energy consumers, DLE might be systematically higher than our central estimate, as they may only be able to access less efficient technologies than the average mixes assumed in EP2050+. This is likely, as low energy consumers are generally also of low income, and hence unlikely to be able to afford devices like heat pumps. Finally, even if there is theoretically space for current (or even increased) levels of inequality, this doesn't imply that these levels are fair, justified, or acceptable. With all this in mind, several alternative messages can thus be gathered from Fig. 5:

First, consider a future where the average Swiss footprint in 2050 is relatively high at ~60–90 GJ cap[-1] – towards the upper end of 1.5 °C and 2 °C scenarios (and slightly below or at the level implicit in EP2050+ NZB). In this case, energy inequality could rise above current levels without preventing the lowest consumers accessing DLE, even for a high DLE estimate of ~30 GJ[5]. Alternatively, if inequality was at or below current levels, the energy footprint of the lowest consumers could rise well above DLE. Critically, this future of high average Swiss energy use can be interpreted in different ways (see the Fig. 5b). If technological deployment matches the highest ambitions imagined in IPCC scenarios, it could be understood as a relatively egalitarian world where all countries have converged to a high, but sustainable per-capita level of energy use. Or if technological progress is less grand, particularly regarding negative emissions, it could be understood as a sustainable world where average global energy use is lower, but global inequalities allow Switzerland to retain a substantial and unjustified energy privilege[21].

Second, consider a future where average energy consumption in Switzerland is much lower, and consistent with a relatively egalitarian world of lower global energy use due to low negative emissions dependence (see Fig. 5b). If average Swiss consumption matches average global energy use in the lowest 2 °C scenario (~36 GJ cap[-1] year[-1]), then DLE of 19.5 GJ cap[-1] year[-1] can be secured for the lowest consumers at slightly higher than current levels of energy inequality, while securing DLE of 30 GJ requires inequality to fall to a Gini of ~0.1 – roughly our estimated fair level. If average Swiss energy use matches the lowest 1.5 °C scenario (~25 GJ cap[-1] year[-1]), energy inequality would have to fall to the same fair level to secure 19.5 GJ of DLE, and even securing a low DLE floor of 15 GJ would require the current

energy Gini to reduce marginally (to ~0.23). All these futures go far beyond that envisaged by EP2050+, but the current results suggest they are possible, in theory, without reducing human wellbeing.

## Discussion

In line with previous work on high-income countries, we find DLE is far lower than current Swiss energy consumption. This is unsurprising, but it is worth fully putting it into context – we estimate that only ~25% of Swiss energy use is currently directed towards meeting human needs, and what remains is a combination of affluence and waste. In the Swiss net-zero scenario we consider, both DLE and broader Swiss energy use fall by ~50% by 2050 due to technological change, so this 25/75 ratio of DLE to waste and affluence remains and 2050 DLE is only ~13% of the current Swiss energy footprint. Furthermore, this is despite our DLE estimate being higher than previous global models[4,5], which we suggest underestimate energy requirements by perhaps 20% due to omitting key categories that we included here (household furnishings and cleaning products; public space for leisure, art, and culture; public administration, security, and research). While not explicitly defined in the original decent living standards work[1,2], these are arguably essential for meeting these living standards.

We also found that DLE can be much higher in certain contexts. For example, the higher mobility requirements of the (diminishing) rural population might mean their DLE is around 25% higher than for urban populations. Further, a single technological factor such as access to a heat pump (rather than a gas boiler), or a modern electric car (rather than an older gasoline car), can reduce total DLE for an individual considerably (by up to 15%). The reduction in DLE for an individual living car-free relative to someone with high car dependence is 13% in 2020, on average (19% for rural inhabitants), but this reduction drops surprisingly low (8%) in 2050. This highlights issues with how technologically focused net-zero scenarios are designed, and our use of such a scenario for sufficiency-based analysis.

The overarching issue is that the high technological ambition assumed in net-zero scenarios can be uneven, and is typically not accompanied by ambitious – or even modest – demand-side changes[10,27,28]. These issues are present in the EP2050+ net-zero scenario that underpins our DLE estimates. For example, EP2050+ does

not assume any substantial shift away from the current modal dominance of car-transport in Switzerland, and to mitigate this, it assumes ambitious reductions in the energy intensity of car transport that are not matched by rail transport. If the energy intensities underpinning our DLE estimates were obtained from a model that assumed a large modal shift to public transport, efficiency improvements may have been greater for those transport modes. DLE estimates for those living without cars would thus have been lower. This highlights the internal inconsistency of our use of technologically focused scenarios from EP2050+ for sufficiency-focused analyses. The inconsistency is lessened as we were not considering a scenario of strict sufficiency as in other work[5,29], but rather using DLE as a consumption floor to explore the space remaining for inequality. And this approach has the advantage of allowing DLE to be based upon the specific Swiss context – thus avoiding the many generic assumptions made in previous DLE work – but without the need for a bespoke energy-economy model. Nonetheless, in a Swiss future focused more centrally upon the sustainable provision of wellbeing, technological progress would be focused in different sectors, and this is something we have not captured.

Inequality is a critical issue that we explore at a relatively high level. Nonetheless, our results provide various insights. First, as the energy requirements of securing wellbeing are so much lower than the average use projected in the EP2050+ net-zero scenario, there is theoretically a large space within this scenario for energy inequality, without compromising access to DLE for the lowest consumers. EP2050+ assumes Swiss domestic energy use halves by 2050, with the average falling to ~50 GJ cap⁻¹, which implies an energy footprint of 85 GJ cap⁻¹ if net imports remain proportionally unchanged. As our central DLE estimate for 2050 is less than 25% of this, current energy inequality could more than double before the lowest consumers were at risk of falling below DLE. However, this is a theoretical result, and it requires both practical and ethical responses.

First, in practice it is neither desirable nor likely that energy inequality would rise until consumption of the lowest consumers fell to DLE. It is undesirable, as our estimate of fair energy inequality – while highly uncertain – is far below the current level, so any increase in inequality would increase this misalignment. It is unlikely, as inequality has been rising very slowly in Switzerland over the past two decades, and even if this rate continued to 2050 energy inequality would only rise marginally. Low carbon transitions do pose risks of increasing inequalities, but these are well known and there are many suggestions for how to make transitions more progressive[30,31].

Second, perhaps most importantly, there are clear arguments that Swiss energy use should be reduced much further than EP2050+ implies. We highlighted above that the energy use EP2050+ NZB assumes in 2050, when scaled-up to include net-imports of embodied energy, is at the upper end of the global averages assumed in IPCC scenarios consistent with 1.5–2 °C of warming. Similarly, the levels of negative emissions assumed would likely be unsustainable if scaled up globally. This Swiss consumption may not in itself be unsustainable if average global consumption remained much lower, but there is no ethical justification for Switzerland to retain this considerable energy privilege[21]. Even domestically, concerns have been raised about the ability of the clean energy capacity assumed in EP2050+ scenarios to meet peak winter demands[32,33], implying that further demand reduction may be necessary if supply cannot be sustainably increased. Finally, note that research in other high-income countries suggests domestic per-capita energy use can be reduced to levels 20% lower than EP2050+ NZB, without compromising quality of life[10].

The current work shows that it is certainly possible to secure decent living conditions for all in Switzerland, and support a large degree of affluence beyond this, even with much more ambitious reductions in Swiss energy use. But this is only feasible if the focus is shifted such that human wellbeing, and the social factors that drive energy demand, become as central a consideration as technological progress.

## Methods

### Wellbeing and decent living standards

Debates around human wellbeing normally cluster around hedonic and eudemonic conceptions[34]. Broadly, questions of individual emotion, happiness, pleasure, and comfort are central to the former, while questions of social conditions, flourishing, purpose, meaning, and functioning are the foci of the latter. Human needs work sits within the latter eudemonic tradition, and the Decent Living Standards (DLS) framework[1] was developed with reference to various theories of needs[35,36]. The primary goal of DLS was to bridge the gap between the abstractions of needs theories and material reality. DLS thus proposes an inventory of material resources that are required for a decent life in our modern industrial societies, which go beyond mere subsistence and the extreme poverty that international income poverty lines measure.

The DLS inventory used in the current work (Table 1) is built upon that used in previous global modelling[4,6], with some significant additions. These include household furnishing (which were previously implicit in the assessment of energy embodied in residential construction, but we consider here explicitly) and cleaning products (which were previously absent). We also add public indoor space, and other public activities, including security and research. These were not included in previous DLS studies, but they are clearly required for meeting human needs like social participation & understanding, safety, healthcare, etc.

We use a mix of three approaches to fully quantify activity-levels across the DLS inventory:

First, where the specific Swiss context is expected to influence required activity-levels (and we can quantify this), we make bespoke estimates. Food and mobility are good examples – we estimate required food consumption using data from the United Nations FAO, which accounts for the population structure of Switzerland; we estimate required mobility per-person by considering demographic and spatial characteristics of Switzerland, as well as current mobility patterns in terms of the trip purposes and modes used.

Second, when there is nothing specific about Switzerland that leads us to believe that more or less consumption is required to meet a particular need (or we simply cannot assess what this variation should be), we use a general assumption made in previous global modelling. Examples are household direct water use (50 L cap⁻¹ year⁻¹) and annual consumption of new clothing (3.6 kg cap⁻¹).

Finally, where it is not possible for us to properly quantify an activity-level, we rely upon more heuristic methods (for example, for Healthcare and for Communication and Information). Full details on activity-levels are in Supplementary Methods 1.

### Decent living energy

DLE models are considered bottom-up, as they estimate the energy requirements of human wellbeing by building up piece-by-piece from a specific set of goods and services – in this case, the decent living standards inventory. In contrast, top-down methods consider society-wide indicators of human wellbeing, such as life expectancy or life satisfaction, and analyse how much energy or resources societies currently use to reach sufficient levels on these indicators[5]. Consequently, they do not consider how energy and resource use is translated by societies into the goods, services, and conditions that contribute to meeting human needs – they are blind to the numerous inefficiencies occurring along this chain and tend to overestimate the environmental impacts of providing human wellbeing. The deficiencies of bottom-up models typically bias them in the opposite direction – they are more likely to omit things that are relevant to human wellbeing than include things that are not, and they typically

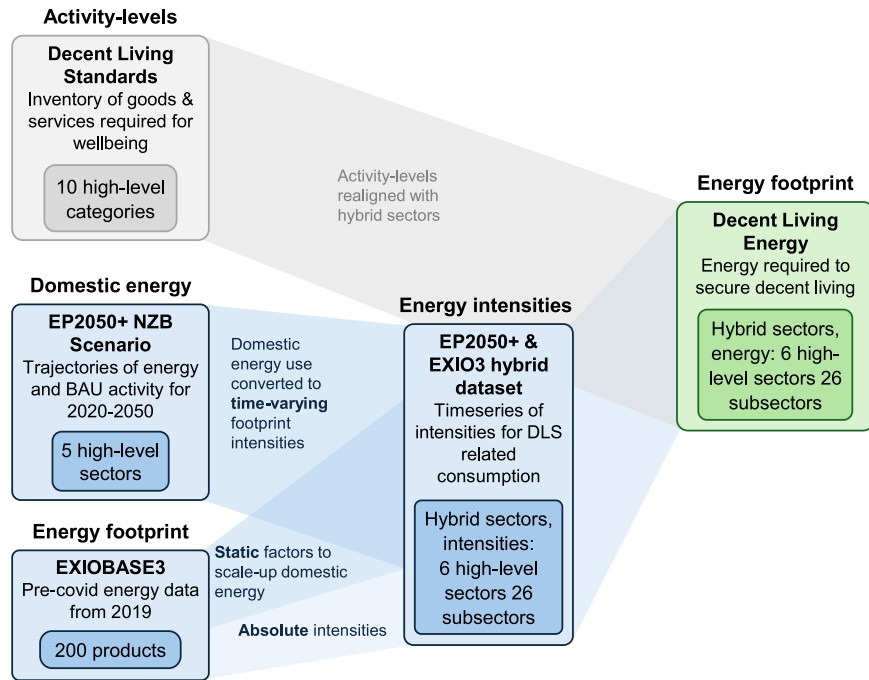

**Fig. 6 | Summary of the methodology for estimating decent living energy.** Activity-levels are defined for each Decent Living Standard, and energy intensities derived from EP2050+ and EXIOBASE3. All calculations were undertaken in a spreadsheet model (MS Excel).

assume that goods and services are translated into human wellbeing with near-perfect efficiency via, for example, unrealistically high distributive equality. The omissions of existing work highlighted in the previous subsection (household furnishing, public indoor space, etc.) represent a good example of this issue of underestimation. Therefore, by broadening the dimensions considered in decent living standards, we aim to mitigate this in the current DLE model. Nonetheless, DLE models, including ours, are best suited to defining ambitious minimum thresholds of energy use that are required to meet basic needs, and secure a decent standard of living well beyond subsistence.

The difficulties of DLE models lie in the process of specifying appropriate activity-levels, and deriving associated energy intensities based on an assumed type and level of technology. But once these activity-levels and intensities are defined, estimating DLE is mathematically simple, and involves multiplying them together and summing across all dimensions of the decent living standards inventory[3,4]. The methodology is summarised in Fig. 6.

**Energy intensity calculations**

Previous DLE work generally derived final energy intensities for each aspect of the DLS inventory by looking to literature from Life Cycle Assessment and Input Output analysis[5,6]. Calculations were largely done bottom-up at the product-level, by estimating, for example, the energy required to produce a particular item of clothing, to wash a single kg of clothing, and to manufacture a washing machine. Work at the global level did this with differing assumptions about technological efficiency, thus producing distinct scenarios[4,5]. Differences in energy intensities between countries were sometimes estimated robustly (for example, for thermal comfort) but other times quite crudely (for example, for construction).

In the current work, we look primarily to a net-zero emissions scenario from the Swiss government – namely, the EP2050+ Net Zero Basis scenario[20] – then to other data measuring Switzerland's current energy footprint so that we include energy used in global supply chains supporting Swiss consumption. This approach has the advantage of producing energy intensities specific to Switzerland, and it allows us to produce a timeseries of DLE matching the horizon of EP2050+. The

main drawback is that the categories used in both datasets are not aligned with the DLS inventory of Table 1 and correcting this is a non-trivial issue (we describe our navigation of this in Supplementary Methods 2). All intensities are summarised in Table 3 and Table 4.

Three different steps are involved when estimating our energy intensities (see Fig. 6):

The first simply uses domestic intensities taken from the EP2050+ Net Zero Basis scenario. This is done where an EP2050+ sector can be directly related to a DLS dimension. For example, EP2050+ reports the final energy intensity of residential heating and the associated floor space (over the 2020–2050 period), so a time-varying intensity can easily be extracted (in MJ m$^{-2}$).

The second step involves using final energy footprint data from EXIOBASE 3[22] to estimate the fraction of energy use that is missed by the domestic scope of EP2050+. For example, EP2050+ projects energy use for Swiss healthcare, but EXIOBASE tells us that domestic energy use only makes up ~40% of the total Swiss healthcare footprint, thus 60% is not covered by EP2050 +. Factors such as this are used to increase the EP2050+ intensities, and we assume the share of the Swiss footprint occurring outside of Switzerland remains fixed from 2020-2050.

The third step is used only where a part of the DLS inventory cannot be aligned with an EP2050+ sector, and hence has not been prescribed an energy intensity via the two steps above. In this case, we use the absolute energy intensities from EXIOBASE, and heuristic estimates of how these may change from 2020 to 2050. Clothing purchases are an example – these are not covered by EP2050+ and are largely imported, but EXIOBASE data can be used to estimate the energy intensity of Swiss consumption (in MJ kg$^{-1}$).

**Domestic energy intensities**

Energy Perspectives 2050 +[20] develops scenarios describing how Switzerland can move to a net-zero emissions energy in 2050. Multiple scenarios are available, and we focus on the central Zero Basis one. This assumes widespread, rapid deployment of well-known mitigation strategies: renewable energy (hydropower, solar, wind, biomass, etc.), energy efficiency (better insulated building envelops, district heating systems, LED lighting, etc.), alternative fuels (hydrogen, biomass, etc.),

**Table 3 | Summary of energy intensities used in the DLE model for energy use within Switzerland**

| Category | Source | 2020 | 2025 | 2030 | 2035 | 2040 | 2045 | 2050 | Units |
|---|---|---|---|---|---|---|---|---|---|
| **Residential** | | | | | | | | | |
| Space heating | EP | 304 | 260 | 215 | 177 | 145 | 119 | 98 | MJ m$^{-2}$ |
| Cooling | EP | 7.3 | 7.1 | 7.3 | 7.9 | 8.8 | 9.4 | 10.1 | MJ m$^{-2}$ |
| Lighting | EP | 7.4 | 4.7 | 4.0 | 3.6 | 3.3 | 3.0 | 2.8 | MJ m$^{-2}$ |
| Appliances | EP | 1,159 | 1,037 | 940 | 885 | 854 | 826 | 798 | MJ cap$^{-1}$ |
| Information technology | EP | 672 | 674 | 670 | 659 | 638 | 615 | 594 | MJ cap$^{-1}$ |
| Water heating | EP | 3419 | 3047 | 2658 | 2311 | 2031 | 1791 | 1584 | MJ cap$^{-1}$ |
| Cooking | EP | 655 | 634 | 608 | 582 | 559 | 544 | 539 | MJ cap$^{-1}$ |
| Clothing | EX-HH | 39 | 37 | 35 | 33 | 32 | 30 | 29 | MJ cap$^{-1}$ |
| Cleaning products | EX-HH | 2.2 | 2.1 | 2.1 | 2.0 | 1.9 | 1.9 | 1.9 | MJ cap$^{-1}$ |
| Household furnishings | Other | 0.010 | 0.009 | 0.009 | 0.009 | 0.008 | 0.008 | 0.007 | MJ m$^{-2}$ |
| Other consumer goods | EX-HH | 156 | 149 | 142 | 136 | 129 | 123 | 117 | MJ cap$^{-1}$ |
| **Transport** | | | | | | | | | |
| Cars and motorbikes | EP | 1.08 | 0.98 | 0.84 | 0.71 | 0.60 | 0.52 | 0.46 | MJ pkm$^{-1}$ |
| Buses | Other | 0.18 | 0.17 | 0.15 | 0.15 | 0.14 | 0.13 | 0.13 | MJ pkm$^{-1}$ |
| Rail | EP | 0.29 | 0.27 | 0.26 | 0.24 | 0.23 | 0.22 | 0.22 | MJ pkm$^{-1}$ |
| Air transport | EP | 1.18 | 1.07 | 0.96 | 0.87 | 0.78 | 0.71 | 0.64 | MJ pkm$^{-1}$ |
| Road transport (beyond direct) | EX-Tot | 0.14 | 0.13 | 0.12 | 0.11 | 0.09 | 0.08 | 0.06 | MJ pkm$^{-1}$ |
| Rail transport (beyond direct) | EX-Tot | 0.32 | 0.29 | 0.27 | 0.24 | 0.20 | 0.17 | 0.14 | MJ pkm$^{-1}$ |
| Air transport (beyond direct) | EX-Tot | 0.30 | 0.29 | 0.27 | 0.26 | 0.24 | 0.23 | 0.21 | MJ pkm$^{-1}$ |
| **Services** | | | | | | | | | |
| Healthcare | EP | 1895 | 1666 | 1509 | 1410 | 1,329 | 1,259 | 1,217 | MJ cap$^{-1}$ |
| Education | EP | 534 | 464 | 414 | 374 | 334 | 309 | 290 | MJ m$^{-2}$ |
| Public administration | EP | 665 | 570 | 519 | 465 | 412 | 382 | 356 | MJ cap$^{-1}$ |
| Trade | EP | 631 | 564 | 507 | 457 | 407 | 371 | 345 | MJ m$^{-2}$ |
| Telecommunication | EP | 1,513 | 1,326 | 1,180 | 1,063 | 961 | 887 | 817 | MJ cap$^{-1}$ |
| Public space | EP | 582 | 514 | 461 | 416 | 370 | 340 | 318 | MJ m$^{-2}$ |
| **Industry** | | | | | | | | | |
| Water supply, waste management | EP | 184 | 136 | 111 | 95 | 83 | 75 | 68 | MJ cap$^{-1}$ |
| Construction | EP | 671 | 638 | 606 | 579 | 558 | 538 | 519 | MJ cap$^{-1}$ |
| Electricity infrastructure | Other | 0.17 | 0.16 | 0.16 | 0.15 | 0.14 | 0.13 | 0.12 | GJ GJ$^{-1}$ |
| **Agriculture** | | | | | | | | | |
| Total | EP | 1068 | 1015 | 949 | 886 | 851 | 819 | 789 | MJ cap$^{-1}$ |
| **Freight** | | | | | | | | | |
| Road (direct) | EP | 2.34 | 2.19 | 1.96 | 1.75 | 1.60 | 1.49 | 1.40 | MJ tkm$^{-1}$ |
| Rail (direct) | EP | 0.17 | 0.16 | 0.15 | 0.14 | 0.14 | 0.13 | 0.13 | MJ tkm$^{-1}$ |
| Road (beyond direct) | EP-EX | 19% | 19% | 20% | 20% | 20% | 20% | 18% | - |
| Rail (beyond direct) | EP-EX | 52% | 52% | 51% | 50% | 47% | 44% | 39% | - |

In the Source column, EP indicates that the data source is Energy Perspectives 2050+; EX-HH and EX-Tot that the data is from EXIOBASE3 (using only household demand and total demand, respectively); EP-EX that EP2050+ data is scaled up using EXIOBASE3; and Other that another data source is used, as detailed in Supplementary Table 6.

and electrification (residential and commercial heat pumps, electric road vehicles, etc.). Alongside this, it assumes substantial deployment of carbon capture and storage. In contrast to these ambitious technological transformations, activity-levels – and hence Swiss living standards – are fixed to business-as-usual levels across all EP2050+ scenarios. While we focus only upon final energy, the assumed deployment of low-carbon technologies in EP2050+ may also imply reduced primary energy consumption in key sectors like transport (which would be the case even with activity-levels fixed).

We extract final energy consumption by sector from the EP2050+ NZB outputs, to be consistent with the DLE methodology. The data are reported for five high-level sectors (residential, services, transport, industry and agriculture) and various subsectors, and we produce a hybrid sector-list to align these with the DLS inventory of Table 1 as closely as possible (see Supplementary Table 6). Much of the industry sector is treated differently, for two reasons: (i) the EP2050+ industrial categories are not final products but rather intermediate ones (steel, cement, glass, etc.) which have no direct relationship with decent living categories, and (ii) most of the goods in the DLS inventory that are consumed in Switzerland (washing machines, cold storage, computers and phones, etc.) are not manufactured domestically, so to consider these we must look to energy footprint data.

An important implication of this approach – that is, our integration of energy intensities from the NZB scenario into our DLE model – is that we implicitly assume the same mix of end-user technologies deployed in the NZB scenario in our DLE estimates. This means that the energy intensities we use for private vehicles, for example, reflect the average Swiss stock assumed in EP2050+, which includes SUVs and other oversized vehicles. The DLS framework is not prescriptive about what private vehicles are and are not decent, so to speak, but it would

**Table 4 | Summary of energy intensities used in the DLE model for imported energy use**

| Category | Source | 2020 | 2025 | 2030 | 2035 | 2040 | 2045 | 2050 | Units |
|---|---|---|---|---|---|---|---|---|---|
| Residential | | | | | | | | | |
| Clothing | EX-HH | 3933 | 3740 | 3557 | 3383 | 3217 | 3059 | 2909 | MJ cap$^{-1}$ |
| Cleaning products | EX-HH | 4550 | 4409 | 4268 | 4126 | 4028 | 3981 | 3934 | MJ cap$^{-1}$ |
| Household furnishings | Other | 19 | 18 | 17 | 16 | 15 | 15 | 14 | MJ m$^{-2}$ |
| Other consumer goods | EX-HH | 513 | 491 | 470 | 448 | 425 | 405 | 385 | MJ cap$^{-1}$ |
| Transport | | | | | | | | | |
| Road transport (beyond direct) | EX-Tot | 0.76 | 0.71 | 0.66 | 0.58 | 0.49 | 0.42 | 0.34 | MJ pkm$^{-1}$ |
| Rail transport (beyond direct) | EX-Tot | 0.50 | 0.46 | 0.43 | 0.38 | 0.32 | 0.27 | 0.22 | MJ pkm$^{-1}$ |
| Air transport (beyond direct) | EX-Tot | 0.44 | 0.42 | 0.40 | 0.38 | 0.35 | 0.33 | 0.31 | MJ pkm$^{-1}$ |
| Services | | | | | | | | | |
| Healthcare | EP-EX | 60% | 60% | 60% | 60% | 60% | 60% | 60% | - |
| Education | EP-EX | 42% | 42% | 42% | 42% | 42% | 42% | 42% | - |
| Telecommunication | EP-EX | 70% | 70% | 70% | 70% | 70% | 70% | 70% | - |
| Public space | EP-EX | 45% | 45% | 45% | 45% | 45% | 45% | 45% | - |
| Public administration, research | EP-EX | 40% | 40% | 40% | 40% | 40% | 40% | 40% | - |
| Trade | EP-EX | 73% | 73% | 73% | 73% | 73% | 73% | 73% | - |
| Industry | | | | | | | | | |
| Water supply, waste management | EP-EX | 11% | 11% | 11% | 11% | 11% | 11% | 11% | - |
| Construction | EP-EX | 66% | 66% | 66% | 66% | 66% | 66% | 66% | - |
| Electricity infrastructure | Other | 0.016 | 0.015 | 0.014 | 0.013 | 0.013 | 0.012 | 0.011 | GJ GJ$^{-1}$ |
| Agriculture | | | | | | | | | |
| Total | EP-EX | 68% | 68% | 68% | 68% | 68% | 68% | 68% | - |
| Freight | | | | | | | | | |
| Road (beyond direct) | EP-EX | 55% | 55% | 57% | 57% | 57% | 56% | 54% | - |
| Rail (beyond direct) | EP-EX | 63% | 63% | 62% | 61% | 58% | 55% | 50% | - |

In the Source column, EX-HH and EX-T indicate that the data source is EXIOBASE3 (using only household demand and total demand, respectively); EP-EX that Energy Perspectives 2050+ data is scaled up using EXIOBASE3; and Other that another data source is used, as detailed in Supplementary Table 6.

be difficult to argue that these oversized vehicles are – despite our implicit inclusion of them. Our DLE estimate should thus be understood as a decomposition of an ambitious net-zero emissions scenario for Switzerland into the energy that is providing DLS, and that supporting consumption in excess of this. It is not a transformative, egalitarian sufficiency scenario, as has recently been explored elsewhere[29].

## Supply chain energy intensities

The energy footprint data we use is from EXIOBASE 3, a global, multiregional environmentally-extended input-output database with 200 product categories across 49 countries/regions[22]. Swiss energy footprint data from EXIOBASE describes the energy embodied in all Swiss consumption – household, government, non-profit institutes, and infrastructure investment – and it associates this energy with all 200 final products and 49 world regions. It describes, for example, the embodied energy of producing all motor vehicles purchased in Switzerland in 2019, and the share of this occurring in Switzerland (4%) and elsewhere (96%). We use the energy extension (Energy Carrier Net Total) describing the primary energy footprint of Switzerland in 2019, and ignore more recent data affected by COVID19. To convert this to a final energy footprint, we simply reduce it uniformly (i.e., across all product categories) by the 2020 ratio of final to primary energy reported in Switzerland by EP2050+, which is 0.75. EXIOBASE data was extracted with the open-source Python package, pymrio (https://doi.org/10.5281/zenodo.1146054).

## Energy inequality

To account for energy inequalities within Switzerland, we implement energy Gini coefficients in the simple analytical formulas recently provided by Pauliuk[26]. By assuming a common distributional form of the Lorenz curve, these relate the minimum DLE requirement for the bottom decile of a population (DLE) to the mean consumption of the full population (avg), and the top decile (max), via the Gini coefficient (G), with all energy data expressed in per capita terms:

$$avg = \frac{1+G}{1-G} \times DLE \quad (1)$$

$$max = avg \times 10 \times 0.1^{\left(\frac{1-G}{1+G}\right)} \quad (2)$$

For example, when G is 0.2 (or 0.5), Eq. (1) suggests that securing DLE for the bottom decile of a population requires average energy consumption of 1.5×DLE (3×DLE), and Eq. (2) that maximum consumption will be roughly 3×DLE (14×DLE).

We assume that the current energy Gini for Switzerland matches the Gini of consumption expenditure, as data suggests that inequality in energy footprints and expenditure are very similar at the level of the latter observed in Switzerland[37]. This expenditure Gini is reported by the Swiss Federal Statistical Office (FSO)[38] most recently in 2014 at 0.26. Other data from the FSO[39] suggests that the Gini of disposable income has risen marginally since then (from 0.29 to 0.3 from 2014 to 2021). We thus assume the expenditure Gini, and hence energy Gini, have risen by the same amount and reached 0.27.

We also make a heuristic estimate of fair inequality following previous work. This took the fair maximum income ratios reported by Kiatpongsan and Norton[40] from survey data in 40 countries, and translated these into fair levels of energy and carbon footprint inequality[41]. Specifically, Swiss residents were found to consider a ratio of ~5:1, on average, to be fair for the income of CEO's and unskilled workers.

The simple parameterisation of inequality we use allows us to easily consider a range of Gini coefficients, but future work would

benefit from empirically analysing the specific distribution of energy inequality in Switzerland.

**Reporting summary**

Further information on research design is available in the Nature Portfolio Reporting Summary linked to this article.

## Data availability

The energy data generated in this study, and model inputs used for the calculations, are provided in the Supplementary Data 1.

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

## Acknowledgements

J.M.H., V.F.R., and S.N. were supported by the SWICE project (Swiss Federal Office of Energy grant); J.M.H. was also supported by the REAL project (ERC ID:101071647); J.M.H., V.R.F., and S.N. are part of the EDITS network, an initiative coordinated by RITE and IIASA and funded by METI, Japan, and S.N. received financial support.

## Author contributions

J.M.H. led the design of the study and method, performed the modelling, and led the writing of the manuscript. V.F.R. and S.N. supported in the design of the study and method, reviewed the modelling results, and contributed to writing the manuscript. EC supported the design of the method and contributed to writing the manuscript.

## Competing interests

The authors declare no competing interests.
