## [Transparent Peer Review file · Nature Communications]

Energy requirements for securing wellbeing in Switzerland and the space for affluence and inequality

Corresponding Author: Dr Joel Millward-Hopkins

Version 0:

Reviewer comments:

Reviewer #1

(Remarks to the Author)
Reviewer report

This paper shows that Switzerland can secure a decent living standard with much lower energy consumption.

This research brings up an important topic on whether we could achieve carbon neutrality from the demand side approaches, while maintaining a decent living standard of humans.

However, some major concerns need to be addressed before publication.

Concept

1.1

A more important question: what is the gap between the current living standard and the minimum energy use for DLS?

I recommend adding more scenarios about the “decent living standard”.

Scenario A: Switzerland maintains the current living standard.

Scenario B: Switzerland increases the living standard defined by the author.

Scenario C: Switzerland maintains the decent living standard defined by the author.

The reason for adding scenarios A and B is that it is more difficult to require people to downgrade their current living standards, even if they are still decent.

Clarifying how the gap between the current life standard and DLS can add more value to this paper.

1.2

More context about energy supply vs energy demand

While reducing energy demand is necessary, it is equally important to consider the source of energy supply.

With abundant renewable energy resources, there is no reason that Global North countries should substantially reduce energy demand.

Reducing energy demand makes sense under two conditions from my perspective:

Limited renewable energy supply, which may be the case in Switzerland.

Energy sources rely on depleted energy resources, or biomass (food security).

Methodology

The method is too thin to fully reveal the key assumptions and details in the paper. Core equations (e.g., DLE, Gini) should be listed even if they are cited from other papers.

2.1

The paper is not clear whether the energy demand sectors in DLS are a subset of the energy demand sectors in the NZB scenario. Therefore, it is not clear to me whether this is an apple-to-apple comparison.

2.2

The paper didn't clearly state the rationale for the efficiency improvements. Any evidence or technology that can enable the improvement in efficiency?

2.3

the increasing share of low-carbon technologies may lead to a lower primary energy consumption in the transportation and industry sectors.

For example, in hard-to-abate sectors, we may need hydrogen or biofuels to replace fossil fuels. Heavy-duty vehicles, shipping and aviation may need hydrogen or biofuels in the transportation sectors. The production and storage of hydrogen or biofuels can be lower than fossil fuels.

Other comments:

Line 29: "One third" should be "one-third"

Line 54-56: Can you provide more background on carbon footprint for general users?

Line 82: This scenario is not mentioned in the previous text. A table is needed to explain the assumptions in the main scenarios used in this paper.

Line 114: I expect more details on efficiency and sufficiency. I believe this is another main contribution of this paper. A figure is necessary to decompose the energy use reduction. Among the reduction in energy use per capita, how much is caused by efficiency improvement, and how much is caused by affluence.

Figure 2: I would expect more information in Figure 2.

A subpanel can be added showing the temporal change in energy use by sector.

Can other factors be quantified? For example, energy consumption for young and old, rich and poor (given a specific Gini coefficient)?

Line 146-156: I recommend summarizing these sensitivity tests into a table. This is a bit lengthy for the result section. And where there are only counterfactual factors about transportation? What about shelter & living?

Line 204 -213: This may be more suitable for the discussion.

Figure 4: More details should be added. I didn't fully understand what the figure means until I read the reference.

In the figure, the author should explicitly explain the Gini coefficient is an independent variable, and that energy use is dependent on the Gini coefficient.

A subpanel showing the difference of energy use between the lowest consumer and the highest consumer would help readers understand the inequality.

The figure captions should clearly explain how the shaded area is calculated and the data source.

Reviewer #2

(Remarks to the Author)

Reviewer #3

(Remarks to the Author)

The manuscript deals with very timely and relevant topics regarding the energy transition and the possibility to ensure decent living standards for the entire world population, while respecting planetary boundaries. Previous research, that some of the authors of the manuscript largely contributed to, already estimated the amount of per capita energy consumption that would be needed to achieve wellbeing and decent living conditions. In this manuscript, the authors aim at performing specific estimates of the amount of energy that would be needed to achieve such conditions for people specifically living in Switzerland. Particularly, they aim at enriching previous estimates of the energy amounts needed to ensure decent living standards (e.g., decent living energy DLE), by accounting for factors about public indoor space, government administration, and research, and by considering how heterogeneity among the population (whether people in rural or urban contexts, use the car or not, use an electric vehicle or an internal combustion one, live in buildings equipped with heat pumps or with gas boilers) may affect such estimates. Having identified the needed amount of decent living energy on varying of such factors, they also aim at exploring the level of inequality that underlies the resulting DLE scenarios, and to clarify the potential for Switzerland to achieve a just energy transition, capable to ensure wellbeing for all its citizens.

I have to say I do not feel completely at ease with the requirement by the journal, which require to first present and discuss the results, and then the methodology used to get to them. Also, the journal has quite strict length constraints, which limit the available space for providing methodological details. Maybe these are among the reasons why I am not fully convinced that the results presented in this manuscript can be entirely understood based on the evidence it offers: I cannot fully grasp how the authors have obtained their results, and I would invite them to more explicitly indicate their assumptions and methodologies. Particularly, I would appreciate if the authors could clarify the following four points.

1. How the specific DLE estimates for Switzerland have been obtained, compared with previous DLE estimates made for

global coverage models. To clarify this, the authors could for instance add a few columns to Table 1 and, for each decent living standard (DLS) that was identified by previous literature (plus the new ones specifically added by the authors), respectively:

- show the estimates of decent living energy (DLE, GJ/cap) that were estimated by previous literature about global models;
- show the DLE estimates the authors specifically made for Switzerland (GJ/cap) via their work;
- and briefly indicate how such DLE estimates for Switzerland were obtained, namely which specific assumptions the authors made when moving from global DLE estimates to specific DLE estimates for Switzerland.

2. Starting from such DLE estimates, that I suppose are those referring to years 2019/2020 (e.g. the current situation), it is not clear to me how DLE values have been estimated for year 2050 (and possibly also for intermediate years, as Figure 1 seems to suggest there is no linear relationship between DLE in 2020 and DLE in 2050). The manuscript indicates that, for this purpose, the estimates produced by the “Energy perspectives 2050+” (EP2050+) scenarios about future domestic energy demand for Switzerland are combined with the EXIOBASE estimates accounting for energy demand associated with good and services that are consumed in Switzerland but produced outside Switzerland. How are these two elements combined with the DLE estimates for years 2019/2020, to produce the DLE estimates for year 2050? Are the DLE estimates for year 2020 re-scaled proportionally to the reduction in energy demand that the EP2050+ assumes between years 2020 and 2050? Overall, I would invite the authors to explicitly explain how they combine the three elements (EP2050+, EXIOBASE, and DLE estimates for 2020) to obtain their DLE estimates for 2050. I am aware this piece of information is most likely reported in the Supplementary materials, but I think this is key to the results presented in the manuscript and therefore recommend it is explicitly indicated in the main body of the manuscript.

3. I am a bit puzzled about some elements being presented as results, while in my understanding they are closer to assumptions made by the authors. For instance, at page 8, from line 260 the authors present results of their models, starting with “We found that DLE can be much higher in certain contexts”, and as an example they report that “the higher mobility requirements of the rural population might mean the DLE is around 25% higher than for the urban population”. From my understanding, these are not results of the model, but assumptions the authors made about the mobility needs by the rural population, compared to the urban one – assumptions that in fact are reported at the top of page 5. So, which are the model results and which are the assumptions the authors gave as inputs to their models? Overall, I have the impression this difference between model inputs/assumptions and model output/results is sometimes blurred and the manuscript would benefit from a clear indication of what the assumptions are and what the results are. Also, whenever the authors present their assumptions, I would appreciate they could justify them by referring to previous literature. This would increase the strength, reliability and trustworthiness of their results.

4. I am sorry but I feel I do not fully understand also the results dealing with inequality. I am familiar with the use of the Gini coefficient to measure income inequality, but in this context it is not clear to me which data the authors use to build an energy related Gini coefficient. Moreover, they state that the Gini coefficient could be increased until 0.4, before the consumers with the lowest energy consumptions are pushed below DLE (lines 200 and 201 at page 7). In my understanding of the Gini coefficient, a higher value means a higher equality, so I would expect that increasing the coefficient would lead to more energy equality allowing a decent energy level for a wider share of the population. As a consequence, the section about energy corridors and especially Figure 4 are not fully clear to me. Overall, also in this case I would invite the authors to more extensively elaborate on their use of the Gini index and its application to their DLE estimates, again providing more details on their methodology and assumptions, to avoid any misunderstandings.

Finally, in the attached PDF document I also added a few minor comments, that I invite the authors to consider. I conclude by thanking them for their attention: I hope my comments are considered helpful and I am looking forward to reading the manuscript in its revised version. I also thank the editors for inviting me to perform such a review.

Best regards,

Francesca Cellina

Version 1:

Reviewer comments:

Reviewer #1

(Remarks to the Author)

The authors have addressed all my concerns.

Reviewer #2

(Remarks to the Author)

Reviewer #3

(Remarks to the Author)

I thank the authors for having thoroughly considered my comments and provided additional information and details in the manuscript to increase my understanding of their work.

The methodology underlying their DLE and DLS estimates is now much clearer to me, as well as the way the elements from the Energy perspective 2050+ study by the Swiss Federal Office of Energy are used. I particularly appreciated the materials the authors reported in the reworked Methods section, as well as the extended description of their approach that is now presented in the Introduction.

I am glad that Table 1 now reports the activity levels estimated by the authors. As I suggested in my previous notes, I would also have appreciated to find there the energy intensities estimated by the authors and the sources from which they were drawn. As these elements, which act as inputs to their model, crucially determine their model's outputs and results, I would have liked to find them directly in the article, rather than in the Supplementary materials where they are currently reported. Thus, if compliant with Nature Communications editorial regulations, I would still suggest the authors to fully report Table 6 directly into the article's main body.

I also thank the authors for having increased the clarity of their analyses on energy inequality and the Gini index. I have to admit that, fully grasping their scope, meaning, and implications was not immediate to me, as they fall out of my expertise. I therefore can only provide limited feedback about the second part of the article.

From my perspective, it might however be useful to also include an explanation of the way the energy Gini for Switzerland was estimated equal to 0.27. Some elements are presented at page 14 (rows 25-30), but to me they are not sufficient to understand why the value is 0.27. I am sure this is due to my limited familiarity with quantitative estimates of inequality and use of the Gini index. However, as this might also be the case of the broad readership of Nature Communications, providing a few additional details might be helpful.

Finally, I am still a bit puzzled by the meaning of the coloured bar reported in Figure 5.b: what is its unit of measurement? Which specific elements of Figure 5.b does it refer to? I guess it has to be interpreted as a qualitative visualisation of a phenomenon that is just related with the elements represented in Figure 5.b. If so, I would suggest the authors to remove it from the figure, which is already quite rich of other visual elements, and to just deal with the related phenomenon in the text. If instead this is not the case, I would invite the authors to a final effort to try to more clearly fit it within the quantitative set-up of figure 5.b.

I conclude by thanking the authors for their attention, and I hope these comments of mine are considered helpful. I also thank the editors for the invitation to perform this review.

Best regards,

Francesca Cellina

Response to reviewers

Many thanks to the reviewers for taking the time to read our manuscript so thoroughly and for the invaluable feedback. We have carefully considered all comments when making our revisions, and we describe our changes in the point-by-point responses below.

Reviewer #1 (Remarks to the Author):

This paper shows that Switzerland can secure a decent living standard with much lower energy consumption. This research brings up an important topic on whether we could achieve carbon neutrality from the demand side approaches, while maintaining a decent living standard of humans. However, some major concerns need to be addressed before publication.

Concept

1.1. A more important question: what is the gap between the current living standard and the minimum energy use for DLS?

I recommend adding more scenarios about the “decent living standard”.

Scenario A: Switzerland maintains the current living standard.

Scenario B: Switzerland increases the living standard defined by the author.

Scenario C: Switzerland maintains the decent living standard defined by the author.

The reason for adding scenarios A and B is that it is more difficult to require people to downgrade their current living standards, even if they are still decent.

Clarifying how the gap between the current life standard and DLS can add more value to this paper.

We agree with the reviewer that it is important to make the gap between current Swiss living standards and DLS much clearer. Scenarios A and C that the reviewer suggests are currently included, with Scenario C represented by our Decent Living Energy scenario, and A by the EP2050+ NZB scenario, which assumes a continuation of current Swiss living standards. The latter was not clear in our previous draft, so we have now clarified this (see Sec 2.1 paragraph 1, and Methods section, ‘Domestic Swiss energy data’ paragraph 1). To this end, we have also added a new figure to highlight these gaps in more detail. Figure 2 (right) clearly illustrates the difference between current activity-levels in Switzerland and decent living standards, for five key areas of consumption – shelter, mobility, food, water and clothing.

On the inclusion of an additional scenario in between decent living standards and current living standards (i.e., Scenario B the reviewer suggests), we do not believe this would add value, for two reasons:

1. First, our final section on energy inequality can be understood precisely to be exploring the Scenario B that the reviewer suggests. Specifically, our inequality analysis takes decent living energy as a minimum level of consumption (i.e., the reviewer’s Scenario A), and considers average levels of energy use above this but below EP2050+ NZB (i.e., the reviewer’s Scenario C). The average living standards being considered here are thus in between decent living standards and current Swiss living standards. So while we do not model a specific, single Scenario B as suggested, we do explore the space within which this scenario would lie.
2. Second, if we were to develop a specific scenario B, the assumed living standards would necessarily be arbitrary, as we have no framework for specifying them. In the absence of such a more rigorous approach, we would likely set activity-levels to be halfway between current levels and decent living levels, which would lead to energy use laying halfway between the associated energy pathways – a result we do not believe would add value.

1.2. More context about energy supply vs energy demand

While reducing energy demand is necessary, it is equally important to consider the source of energy supply. With abundant renewable energy resources, there is no reason that Global North countries should substantially reduce energy demand. Reducing energy demand makes sense under two conditions from my perspective:

1. Limited renewable energy supply, which may be the case in Switzerland.
2. Energy sources rely on depleted energy resources, or biomass (food security).

We agree that the case for further reducing the Swiss energy footprint should be better laid out. We have now done this at various points in the manuscript, including the introduction (paragraph 4), section 2.4.1 (paragraph 1), and the conclusions (2nd to last paragraph).

A key part of this argument rests on the fact that we are considering the full Swiss energy footprint – all the energy use required to produce all goods and services consumed in Switzerland, irrespective of where in the world this production occurs. This implies that the case for reducing the Swiss energy footprint does not just rest on considerations of clean energy availability in Switzerland itself, but on global limits to sustainable energy production – we consider these limits in the inequality analysis of section 2.4.

Another reason for further reducing Swiss energy use is that, like many IPCC scenarios, the technological ambitions underpinning the EP2050+ net zero pathways for Switzerland may prove difficult to meet, particularly the large scale of carbon capture. Similarly, recent research has cast doubt on the ability of the energy systems assumed in these net zero futures to handle peak winter demands. Concerns such as these highlight how energy-sufficiency strategies, which further reduce energy demand, may mitigate high reliance on potentially infeasible technological deployment. We have integrated these points into section 2.4.1 (paragraph 1) and the conclusions (2nd to last paragraph).

Finally, we would argue that estimating the minimum energy requirements of decent living standards is a useful theoretical exercise in itself, irrespective of ecological limits. It is useful for the same reason it's useful to set poverty lines in countries whose average incomes' are far beyond these lines.

Methodology

The method is too thin to fully reveal the key assumptions and details in the paper. Core equations (e.g., DLE, Gini) should be listed even if they are cited from other papers.

We agree, and in particular our consideration of energy inequality was not explained sufficiently. We have expanded the methods (final section, 'Energy inequality') to properly explain the Gini coefficient, and our use of it in the analysis of section 2.4. We also add the core equations relating DLE, via this Gini coefficient, to the mean and max energy consumption of a population.

We have also significantly expanded the first three sections of the methods so as to fully describe the derivation of Decent Living Energy. This involves more detailed descriptions of both activity-levels and energy intensities, and their derivation. The core equation for DLE is then a simple multiplication and summation of activity-levels and energy-intensities, which we clarify at the end of the 'Decent Living Energy' section of the methods.

2.1. The paper is not clear whether the energy demand sectors in DLS are a subset of the energy demand sectors in the NZB scenario. Therefore, it is not clear to me whether this is an apple-to-apple comparison.

We agree this was not sufficiently clear. The mapping of DLS sectors to EP2050+ and EXIOBASE is fully described in the Supplementary Information, but this was not made clear in the main text. We have thus added an additional figure that illustrates the methodology and how, at a high-level, the different databases are linked.

Note also that, as we scale the NZB scenario up to measure the Swiss final energy footprint, and DLS is also measured as a final energy footprint, a direct comparison of these is appropriate. What would not be appropriate is to compare the domestic NZB data only with a DLE footprint estimate, so we refrain from doing so.

2.2. The paper didn't clearly state the rationale for the efficiency improvements. Any evidence or technology that can enable the improvement in efficiency?

Primarily, our energy efficiency assumptions are taken directly from those that underlie the NZB scenario, which are themselves derived from a detailed technological model of the Swiss energy economy. We have now elaborated upon these assumptions at two different points in the paper – section 2.1, and the methods ('Domestic Swiss energy data' subsection).

2.3. The author didn't clarify the difference between the final energy and primary energy use.

The paper only discussed the final energy consumption, but didn't discuss primary energy consumption. While improving energy efficiency can reduce the final energy consumption, the increasing share of low-carbon technologies may lead to a lower energy transition in the transportation and industry sectors.

For example, in hard-to-abate sectors, we may need hydrogen or biofuels to replace fossil fuels. Heavy-duty vehicles, shipping and aviation may need hydrogen or biofuels in the transportation sectors. The production and storage of hydrogen or biofuels can be lower than fossil fuels.

All the existing DLE work we are aware of is focused upon final energy, so to be consistent with this existing literature we also consider only on final energy. This focus is because the literature is interested in how much energy is needed to support decent living standards, but not the losses that happen upstream. We have now clarified this in the Introduction (paragraph 5).

Note also that EP2050+ does fully consider energy supply and assumes a small amount of hydrogen and biofuels in multiple sectors, so these technologies are implicitly accounted for in our results, and we now mention this in the Methods ('Domestic Swiss energy data' subsection).

Other comments:

Line 29: "One third" should be "one-third"

We have actually removed this sentence to cut down the word count, as it was repetitive.

Line 54-56: Can you provide more background on carbon footprint for general users?

We have now defined these in the following paragraph, where we also define the energy footprint measure that we then use throughout.

Line 82: This scenario is not mentioned in the previous text. A table is needed to explain the assumptions in the main scenarios used in this paper.

Although we do not include a specific table, we have now described the EP2050+ scenario we use more thoroughly, in response to other reviewer comments.

Line 114: I expect more details on efficiency and sufficiency. I believe this is another main contribution of this paper. A figure is necessary to decompose the energy use reduction. Among the reduction in energy use per capita, how much is caused by efficiency improvement, and how much is caused by affluence.

We have now more clearly described our results in terms of sufficiency and efficiency (section 2.2) and added a figure illustrating this decomposition (Fig. 2a), as the reviewer suggests.

Figure 2: I would expect more information in Figure 2.

A subpanel can be added showing the temporal change in energy use by sector. Can other factors be quantified? For example, energy consumption for young and old, rich and poor (given a specific Gini coefficient)?

We have now expanded the results shown in Figure 2 (now Figure 3) to show the average annual reductions in energy use for each consumption sector. We have also added the DLE estimate for Switzerland from previous work, following reviewer #2's request. We have not added estimates here for the young and old, because we have not made these estimates as it goes beyond the scope of our current work (indeed this is an important question future DLE should address). However, we have now indicated the range in energy use between the highest and lowest consumers on the new Figure 5a.

Line 146-156: I recommend summarizing these sensitivity tests into a table. This is a bit lengthy for the result section. And where there are only counterfactual factors about transportation? What about shelter & living?

We agree this information would be better summarised in a table, so we have done so in Table 2. We chose to explore two factors for transportation, and only one for shelter & living, as the former is the largest contributor to total DLE in Switzerland.

Line 204 -213: This may be more suitable for the discussion.

We agree that this information would also be appropriate in the discussion section. However, this paragraph provides necessary context for the two paragraphs that follow it, so if it were to be moved, these following paragraphs would also have to be moved. This would leave the discussion much longer, and section 2.4 much shorter, which is not ideal, so for now, we have left these paragraphs in section 2.4. We are happy to reconsider this if the reviewer believes it to be important.

Figure 4: More details should be added.

I didn't fully understand what the figure means until I read the reference. In the figure, the author should explicitly explain the Gini coefficient is an independent variable, and that energy use is dependent on the Gini coefficient. A subpanel showing the difference of energy use between the lowest consumer and the highest consumer would help readers understand the inequality. The figure captions should clearly explain how the shaded area is calculated and the data source.

We agree, Figure 4 was difficult to digest. We have edited Section 2.4 so the figure and underlying calculations are described far more clearly. And we have edited the left panel of the figure (now Figure 5a) to highlight the range in energy use between the lowest and highest consumers, as a function of the Gini.

Reviewer #2 (Remarks to the Author):

Reviewer #3 (Remarks to the Author):

The manuscript deals with very timely and relevant topics regarding the energy transition and the possibility to ensure decent living standards for the entire world population, while respecting planetary boundaries. Previous research, that some of the authors of the manuscript largely contributed to, already estimated the amount of per capita energy consumption that would be needed to achieve wellbeing and decent living conditions. In this manuscript, the authors aim at performing specific estimates of the amount of energy that would be needed to achieve such conditions for people specifically living in Switzerland. Particularly, they

aim at enriching previous estimates of the energy amounts needed to ensure decent living standards (e.g., decent living energy DLE), by accounting for factors about public indoor space, government administration, and research, and by considering how heterogeneity among the population (whether people in rural or urban contexts, use the car or not, use an electric vehicle or an internal combustion one, live in buildings equipped with heat pumps or with gas boilers) may affect such estimates. Having identified the needed amount of decent living energy on varying of such factors, they also aim at exploring the level of inequality that underlies the resulting DLE scenarios, and to clarify the potential for Switzerland to achieve a just energy transition, capable to ensure wellbeing for all its citizens.

I have to say I do not feel completely at ease with the requirement by the journal, which require to first present and discuss the results, and then the methodology used to get to them. Also, the journal has quite strict length constraints, which limit the available space for providing methodological details. Maybe these are among the reasons why I am not fully convinced that the results presented in this manuscript can be entirely understood based on the evidence it offers: I cannot fully grasp how the authors have obtained their results, and I would invite them to more explicitly indicate their assumptions and methodologies. Particularly, I would appreciate if the authors could clarify the following four points.

1. How the specific DLE estimates for Switzerland have been obtained, compared with previous DLE estimates made for global coverage models. To clarify this, the authors could for instance add a few columns to Table 1 and, for each decent living standard (DLS) that was identified by previous literature (plus the new ones specifically added by the authors), respectively:

- show the estimates of decent living energy (DLE, GJ/cap) that were estimated by previous literature about global models;
- show the DLE estimates the authors specifically made for Switzerland (GJ/cap) via their work;
- and briefly indicate how such DLE estimates for Switzerland were obtained, namely which specific assumptions the authors made when moving from global DLE estimates to specific DLE estimates for Switzerland.

We agree, our previous manuscript did not adequately describe our process of estimating DLE in Switzerland and specifically how this process relates to previous global models. Instead of adding these to Table 1 of the Introduction, we have modified Figure 2 and Figure 3 to include DLE estimates from previous global models that are most comparable with our results (Figure 3 allows for a comparison by sector). To Table 1, we have added a column summarising the activity-levels we use to represent decent living standards in Switzerland, distinguishing specific Swiss estimates from generic (and global) normative assumptions. Finally, we have also edited and expanded all discussion of our method. This includes a brief, but clear, overview of the approach in the Introduction (see 2nd to last paragraph).

2. Starting from such DLE estimates, that I suppose are those referring to years 2019/2020 (e.g. the current situation), it is not clear to me how DLE values have been estimated for year 2050 (and possibly also for intermediate years, as Figure 1 seems to suggest there is no linear relationship between DLE in 2020 and DLE in 2050). The manuscript indicates that, for this purpose, the estimates produced by the “Energy perspectives 2050+” (EP2050+) scenarios about future domestic energy demand for Switzerland are combined with the EXIOBASE estimates accounting for energy demand associated with good and services that are consumed in Switzerland but produced outside Switzerland. How are these two elements combined with the DLE estimates for years 2019/2020, to produce the DLE estimates for year 2050? Are the DLE estimates for year 2020 re-scaled proportionally to the reduction in energy demand that the EP2050+ assumes between years 2020 and 2050? Overall, I would invite the authors to explicitly explain how they combine the three elements (EP2050+, EXIOBASE, and DLE estimates for 2020) to obtain their DLE estimates for 2050. I am aware this piece of information is most likely reported in the Supplementary materials, but I think this is key to the results presented in the manuscript and therefore recommend it is explicitly indicated in the main body of the manuscript.

The reviewer’s assumption here is correct – our 2050 DLE estimates are obtained primarily by considering the energy use reduction within EP2050+. We have substantially edited the Methods to make this clear (see ‘Energy intensities’ section) and have also clarified the point briefly in section 2.1 (final paragraph). Finally,

we have also added Figure 6 to the Methods section, which visually summarises our approach and clarifies how EP2050+, EXIOBASE and Decent Living Standards are combined to obtain our DLE estimate.

3. I am a bit puzzled about some elements being presented as results, while in my understanding they are closer to assumptions made by the authors. For instance, at page 8, from line 260 the authors present results of their models, starting with “We found that DLE can be much higher in certain contexts”, and as an example they report that “the higher mobility requirements of the rural population might mean the DLE is around 25% higher than for the urban population”. From my understanding, these are not results of the model, but assumptions the authors made about the mobility needs by the rural population, compared to the urban one – assumptions that in fact are reported at the top of page 5. So, which are the model results and which are the assumptions the authors gave as inputs to their models? Overall, I have the impression this difference between model inputs/assumptions and model output/results is sometimes blurred and the manuscript would benefit from a clear indication of what the assumptions are and what the results are. Also, whenever the authors present their assumptions, I would appreciate they could justify them by referring to previous literature. This would increase the strength, reliability and trustworthiness of their results.

We agree that the difference between results and assumptions is not always clear, so we have attempted to resolve this in the revised text, particularly in various figure legends. Regarding the specific example of mobility, this we would say is a result. It is true that the differences are a direct consequence of the higher levels of mobility we argue are required for decent living in rural areas. But these higher levels are not just an assumption, but rather a result of our analysis of both the specific travel purposes and needs in Switzerland, and the spatial organisation and accessibility of services.

4. I am sorry but I feel I do not fully understand also the results dealing with inequality. I am familiar with the use of the Gini coefficient to measure income inequality, but in this context it is not clear to me which data the authors use to build an energy related Gini coefficient. Moreover, they state that the Gini coefficient could be increased until 0.4, before the consumers with the lowest energy consumptions are pushed below DLE (lines 200 and 201 at page 7). In my understanding of the Gini coefficient, a higher value means a higher equality, so I would expect that increasing the coefficient would lead to more energy equality allowing a decent energy level for a wider share of the population. As a consequence, the section about energy corridors and especially Figure 4 are not fully clear to me. Overall, also in this case I would invite the authors to more extensively elaborate on their use of the Gini index and its application to their DLE estimates, again providing more details on their methodology and assumptions, to avoid any misunderstandings.

We agree that the inequality analysis had not been sufficiently explained. Consequently, we have significantly edited the description in the analysis section itself (Section 2.4), by splitting it into three subsections: the first discussing potentially sustainable total levels of Swiss-energy use; the second how we model inequality and estimate current levels; the third presenting our results. Included here is an explanation of the Gini coefficient (which varies from 0 to 1, where 0 is perfect equality and higher values represent higher inequality).

The left panel of Figure 4 (now Figure 5a) has also been edited to show the relationship between the Gini coefficient, DLE, and average and maximum energy use from our parameterisation. We note clearly that this is not results, but it serves to help understand the results on Figure 5b.

Finally, a full explanation of our parameterisation of inequality is now included in the Methods (‘Energy inequality’ subsection), where equations can also be found in response to reviewer #1’s feedback.

Finally, in the attached PDF document I also added a few minor comments, that I invite the authors to consider. I conclude by thanking them for their attention: I hope my comments are considered helpful and I am looking forward to reading the manuscript in its revised version. I also thank the editors for inviting me to perform such a review.

Best regards,
Francesca Cellina
Senior researcher at the University of applied sciences and arts of Southern Switzerland

Many thanks for these final comments, we hope we have addressed them all in our revision, particularly we hope that the above-mentioned changes to the inequality results and methodological descriptions address the associated points in the PDF.

The only response we should note is regarding our use of the phrase ‘contextual factors’ to describe the scenarios of section 2.3. We would rather retain this generic terminology, then using ‘individual factors’, as we do not want to suggest that the factors we modify are determined by individual choice. They may be of course – if someone has the money for a heat-pump but chooses to spend this on something else, or is well-served by public transport but still predominantly drives. But there could instead be issues of car-dependency, or a lack of sufficient finances for a new heating system or electric vehicle. We feel ‘contextual’ factors is thus more appropriate, as it does not cast a judgement regarding causation.

Our thanks once again for this extremely useful review.

Additional response for both reviewers:

While undertaking our revisions, we realised we had not interpreted the EXIOBASE energy extension correctly – what we thought was final energy use is actually primary energy. We have thus now scaled this down to a final energy footprint estimate. While this has not in any way changed the messages of our conclusions, it does slightly affect our results – most importantly, the ratio of the Swiss energy footprint to domestic Swiss energy use is now ~1.7 (rather than ~2.2), while our DLE estimates have all decreased 5-10%. We have thus updated the figures and text to reflect these updates.

Note that we have checked our updated final energy footprint with other published estimates and the data compares well (e.g., <https://doi.org/10.1016/j.apenergy.2018.01.044> report a ratio of ~1.8 for footprint-to-domestic final energy use in Switzerland).

Response to reviewers

Many thanks to the reviewers for again taking the time to read our manuscript and for the additional feedback. We have considered all these comments when making our revisions, and our changes are detailed below.

Reviewer #1 outstanding comment highlighted by the editor:

Referee #1 corrected the comment of:

“2.3. The author didn’t clarify the difference between the final energy and primary energy use. The paper only discussed the final energy consumption, but didn’t discuss primary energy consumption. While improving energy efficiency can reduce the final energy consumption, the increasing share of low-carbon technologies may lead to a lower energy transition in the transportation and industry sectors”

to:

“the increasing share of low-carbon technologies may lead to a lower primary energy consumption in the transportation and industry sectors”

in their review report, please also address this in the latest responses letter.

Our previous response to reviewer #1 remains applicable here, namely:

All the existing DLE work we are aware of is focused upon final energy, so to be consistent with this existing literature we also consider only final energy. This focus in the literature is because DLE work is interested in how much energy is needed to support decent living standards, and not the losses that happen upstream. We have now clarified this in the Introduction (Page 3, lines 16-19).

Note also that EP2050+ does fully consider energy supply and assumes a small amount of hydrogen and biofuels in multiple sectors, so these technologies are implicitly accounted for in our results, and we now mention this in the Methods (page 15, *Domestic Swiss energy data* subsection).

However, to better respond to the reviewer’s comment, we have now added a clarification to the methods to ensure that their point is recognised in the manuscript (see page 16, lines 5-7).

Reviewer #3 (Remarks to the Author):

I thank the authors for having thoroughly considered my comments and provided additional information and details in the manuscript to increase my understanding of their work.

The methodology underlying their DLE and DLS estimates is now much clearer to me, as well as the way the elements from the Energy perspective 2050+ study by the Swiss Federal Office of Energy are used. I particularly appreciated the materials the authors reported in the reworked Methods section, as well as the extended description of their approach that is now presented in the Introduction.

Many thanks to the reviewer for their positive comments, we are glad to hear our revisions have brought this clarity.

I am glad that Table 1 now reports the activity levels estimated by the authors. As I suggested in my previous notes, I would also have appreciated to find there the energy intensities estimated by the authors and the sources from which they were drawn. As these elements, which act as inputs to their model, crucially

determine their model's outputs and results, I would have liked to find them directly in the article, rather than in the Supplementary materials where they are currently reported. Thus, if compliant with Nature Communications editorial regulations, I would still suggest the authors to fully report Table 6 directly into the article's main body.

We are happy to do this and have thus brought Supplementary Table 6 into the main text – specifically into the Methods section where energy intensities are described. In order to make this very large table suitable for the main text, we have removed the 'notes' column (which references only minor data sources) and split the table into two – summarising domestic and imported energy intensities separately (in tables 3 and 4, respectively). We have thus maintained the original table in the supplementary information as well.

I also thank the authors for having increased the clarity of their analyses on energy inequality and the Gini index. I have to admit that, fully grasping their scope, meaning, and implications was not immediate to me, as they fall out of my expertise. I therefore can only provide limited feedback about the second part of the article.

From my perspective, it might however be useful to also include an explanation of the way the energy Gini for Switzerland was estimated equal to 0.27. Some elements are presented at page 14 (rows 25-30), but to me they are not sufficient to understand why the value is 0.27. I am sure this is due to my limited familiarity with quantitative estimates of inequality and use of the Gini index. However, as this might also be the case of the broad readership of Nature Communications, providing a few additional details might be helpful.

We agree this should be clarified further. Our assumptions are simple, and we have added a short standalone paragraph to the Methods explaining them (see page 17, lines 18-23).

Finally, I am still a bit puzzled by the meaning of the coloured bar reported in Figure 5.b: what is its unit of measurement? Which specific elements of Figure 5.b does it refer to? I guess it has to be interpreted as a qualitative visualisation of a phenomenon that is just related with the elements represented in Figure 5.b. If so, I would suggest the authors to remove it from the figure, which is already quite rich of other visual elements, and to just deal with the related phenomenon in the text. If instead this is not the case, I would invite the authors to a final effort to try to more clearly fit it within the quantitative set-up of figure 5.b.

I conclude by thanking the authors for their attention, and I hope these comments of mine are considered helpful. I also thank the editors for the invitation to perform this review.

We agree with the reviewer that Figure 5b was rather rich in details, and the addition of the colourbar merely added confusion. We have thus edited the figure significantly, by pulling out the aspects specific to the colourbar from the plot (namely the energy-temperature ranges), and aligning these directly with the colourbar. We have also simplified the colourbar so the descriptive labels align exactly with the discussion in the main text (see page 9 line 16 to page 10 line 8). Finally, we have simplified the colour scheme so there is no misleading correspondence between the colours of the colourbar and those in the plots 5a and 5b.